# "The impact of the Little Orange Book on how parents/carers manage symptoms of illness in children: A mixed methods study"

Amy Johnson[1‡], Kathryn Carruthers[2☉], Matthew Breckons[3☉], Lynette Harland Shotton[4*]

1 Department of Nursing, Midwifery and Health, Northumbria University, Newcastle Upon Tyne, United Kingdom, 2 Department of Health and Social Care, Teesside University, Middlesbrough, United Kingdom, 3 Faculty of Medical Sciences, Newcastle University, Newcastle Upon Tyne, United Kingdom, 4 Department of Social Work, Education and Community Wellbeing, Northumbria University, Newcastle Upon Tyne, United Kingdom

☉ These authors contributed equally to this work.
‡ AJ is first author on this work.
* lynette.shotton@northumbria.ac.uk

**Data Availability Statement:** The ethical approval granted for this study did not include informed consent from participants to share full data sets.

## Abstract

### Background

Previous literature has highlighted the complexity of supporting an acutely unwell child and the unnecessary use of services by parents/carers. The Little Orange Book (LOB) was developed as an information resource for parents/carers of young children to assist in managing symptoms of childhood illness and to encourage the appropriate use of healthcare services.

### Objectives

This study aimed to understand parent/carer views and experiences using the Little Orange Book. Specifically, this study focused on barriers and facilitators to use, the impact on parents' behaviour and the views on improving the LOB.

### Methods

An explanatory sequential mixed-methods approach, including qualitative and quantitative components, was used to collect data regarding parent's experiences using the LOB. Parents and carers in the North-East of England were invited to participate in an online survey and a subset of these respondents took part in qualitative interviews. One-hundred-and-twenty-eight parents completed the online survey and 16 took part in interviews. Qualitative data were analysed using thematic analysis and quantitative data analysed using descriptive statistical analytical methods.

### Results

Three themes were identified within the data: *Increasing parental empowerment in managing their child's health*, *Equality of Access to Health-based Literature* and *Barriers and Facilitators to using the Little Orange Book*.

Therefore, data sets associated with this evaluation will be provided subject to reasonable request by emailing the corresponding author lynette.shotton@northumbria.ac.uk or via the institutional research team as.researchlink@northumbria.ac.uk.

**Funding:** Grant Number Agresso Reference CRP01901 L H Shotton – grant holder Funder - Newcastle Gateshead CCG, Riverside House, Goldcrest Way, Newburn Riverside (Business Park), Newcastle upon Tyne NE15 8NY https://nenc-newcastlegateshead.icb.nhs.uk A J received salary The funders were involved in shaping the survey questions but did not have any involvement in data collection and analysis, decision to publish or preparation of this manuscript.

**Competing interests:** No authors have competing interests

## Conclusions

The Little Orange Book was seen as a valuable form of information to support parents and carers in healthcare decisions. Further development should ensure inclusivity, widen access and view provision of the information as an opportunity for healthcare professionals to engage with parents.

## Introduction

For parents/carers, making decisions about when and where to access healthcare or health services for their acutely unwell child is complex [1]. Following interviews with parents of young children [2], Conlon et al. describe a complex decision-making process in which parents sought care for their children when a 'threshold of capacity for self-management' was exceeded. A recent systematic review suggested that pre-disposing factors (socioeconomic status, ethnicity and race) may influence parental decisions on the use of care, which are often based on perceptions of the urgency of the condition [3]. This review highlighted multiple factors featured within parental decision-making, including a need for reassurance as well as judgments regarding the availability and convenience of services, relationships with General Practitioners and perceptions of the quality of care in emergency departments.

Prompted by dramatic reductions in the use of children's health services during the Covid-19 pandemic [4, 5], research has identified several factors feeding into parental decision-making, including making sense of risk and understanding information about health service availability [6]. While health information has the potential to cause anxiety [7], some suggest that studies examining health service use focus too much on help-seeking behaviours rather than the examination of sense-making, which can create unhelpful, polarised views which classify service use as appropriate or not [8]. Several studies have concluded that resources to support decision-making or increasing parental knowledge of health conditions, including self-care, may play an important role in ensuring appropriate healthcare use [1, 9, 10].

The Little Orange Book (LOB) [11] is a paediatric health resource developed by the NHS Newcastle Gateshead Clinical Commissioning Group (NGCCG) in collaboration with healthcare professionals and parents/carers. The resource aims to support parents/carers with children under 5 years old to manage common childhood illnesses, emergencies, and critical conditions requiring urgent medical attention. The paper version of the LOB is freely available from clinical and community settings based in Newcastle and Gateshead in the Northeast of England and can be downloaded as a digital copy. An initial evaluation investigated the perceived value of the LOB by parent/carers and healthcare professionals, and the impact of its use on unnecessary use of health services [9]. Parent/carers completed face-to-face questionnaires or brief feedback cards and healthcare professionals completed online or paper questionnaires. Participants viewed the layout and content of the LOB positively. The majority of parents/carers reported referring to the LOB in the event of their child being unwell, noting an increase in confidence in self-care and appropriate use of health services. Whilst this evaluation was valuable, there was scope for a further in-depth evaluation focusing on individual parent's experiences of using the LOB and in particular, barriers and facilitators to use.

This study aimed to explore parent/carers' views and experiences using the LOB. Building on the previous evaluation, this study utilised a mixed-method approach and explored in-depth the barriers and facilitators to using the LOB, the perceived impact on parent behaviour when making health-related decisions for their child, and suggestions for improving the

resource. This article outlines the key findings of a commissioned evaluation of the LOB and will make a new contribution to knowledge of parental decision-making, distribution of and access to health information, and considerations for future evaluation methods and research.

## Methods

### Design

This study used an explanatory sequential mixed methods design (containing quantitative and qualitative stages) across two phases [10] underpinned by a pragmatic approach focussed on understanding the implications for future information provision. Phase one was an online survey aimed at understanding broad views and impacts of the LOB. These initial findings informed the second phase, both through allowing purposive sampling based on demographics as well as allowing trends in data to be explored further through semi-structured interviews with parents/carers in order to gain further understanding of the views and experiences of using the LOB. While the survey explored the views of those who had previously received the LOB and those who had not, this paper will focus on parents/carers who received the LOB across both phases, the views of those who did not receive the LOB are presented elsewhere [12].

### Ethics

Ethical approval to conduct the evaluation was obtained from Northumbria University Health and Life Sciences Ethics Committee (reference number: 41385.).

The research team provided information about the study to those who had expressed an interest in taking part and were available to provide further information or answer any questions. Those who participated in the study provided informed consent virtually (online survey) and written (individual/group interview). Verbal consent was also obtained at the beginning of each interview. All participants were provided with a unique identifier for anonymity and no names were included with any data. Regarding the qualitative components, any possible identifying information (e.g. location or names) were removed during the transcription process.

Survey respondents had the opportunity to be entered into a prize draw to win £25 voucher and parents/carers who participated in an individual or group interview received a £25 voucher at completion.

### Participants

A convenience sample of parents/carers from across Newcastle and Gateshead were invited to participate in both phases of the project. The online survey was publicised via an electronic flier shared with over 300 health and community services/groups for families and children, 193 nurseries, and via social media groups, organisational websites, and GP bulletins. A Facebook advertisement was developed and cross-posted via Instagram and Facebook Messenger. Survey respondents had the opportunity to 'opt in' to an interview to further explore their views and experiences of using the LOB.

The inclusion criteria for the study included being over 18 years old, being a parent/carer or guardian (including grandparents), live or lived in the North East of England, and the capacity to give informed consent. In total, 128 individuals completed the online survey. Of these 82 had received and used the LOB, 24 had received and not used the LOB, and 22 had not received the LOB (see Table 1).

**Table 1. Demographics of survey respondents (N = 128).**

| Variable | Received and used the Little Orange Book (N = 82) | Received and did not use the Little Orange Book (N = 24) |
|---|---|---|
| **Age** | 25–56 years (mean = 35.85, SD = 5.12) | 24–50 years (mean = 36.96, SD = 6.49) |
| **Gender** | | |
| Female | 95.12% | 100% |
| Male | 4.88% | 0% |
| **Ethnicity** | | |
| White | 93.90% | 91.67% |
| Black/African/Caribbean/Black British | 3.66% | 0% |
| Asian/Asian British | 1.22% | 4.17% |
| Mixed/Multiple Ethnic Groups | 1.22% | 4.17% |
| **Marital Status** | | |
| Married/Civil Partnership | 92.68% | 91.67% |
| Single | 6.10% | 8.33% |
| Separated | 1.22% | 0% |
| **Number of Children** | | |
| First-time parent | 48.7% | 41.67% |
| Two Children | 72.50% | 69.23% |
| Three Children | 22.50% | 23.08% |
| Four Children | 2.50% | 7.69% |
| More than Four Children | 2.50% | 0% |
| **Employment Status** | | |
| Employed full-time | 43.90% | 54.17% |
| Employed part-time | 36.59% | 33.33% |
| Homemaker/Housewife | 10.98% | 4.17% |
| Self-employed | 3.66% | 0% |
| Carer | 1.22% | 4.17% |
| Unemployed and not currently looking for work | 1.22% | 0% |
| Unable to work | 1.22% | 0% |
| Prefer not to say | 1.22% | 0% |
| Other | | 4.17% |
| **Version of the LOB** | | |
| Paper-based copy | 56.10% | 70.83% |
| Online copy | 10.98% | 10.93% |
| Both online and paper copies | 32.93% | 8.33% |

For phase two, sixteen parents/carers took part in individual and group interviews over Microsoft Teams (n = 13); face-to-face (n = 1), and over the phone (n = 2) (see Table 2).

This paper presents the views and experiences of the 106 survey recipients and 16 interview participants who received a version of the LOB.

## Phase one: Online survey

The online survey was produced using JISC online surveys and was available for completion between 9th April-23rd July 2022. The survey was developed specifically for this study and was created in collaboration with members of the NGCCG.

Initial pages of the survey included information about the study and required digital confirmation of informed consent. The survey included open-text and multiple-choice response options. Survey respondents were directed to one of three pathways based on their prior

**Table 2. Interview participant demographics (N = 16).**

| Variable | |
|---|---|
| **Age** | 25–43 years (mean = 35.19, SD = 4.46) |
| **Use of the Little Orange Book** | |
| Received and used the LOB | 87.5% |
| Received and not used the LOB | 6.25% |
| Had not received the LOB | 6.25% |
| **Ethnicity** | |
| White | 87.50% |
| Black/African/Black British | 6.25% |
| Asian/Asian British | 6.25% |
| **Marital Status** | |
| Married/Civil Partnership | 93.75% |
| Single | 6.25% |
| **Number of Children** | |
| One child | 50.00% |
| Two Children | 50.00% |
| **Employment Status** | |
| Employed full-time | 50.00% |
| Employed part-time | 37.50% |
| Homemaker/Housewife | 6.25% |
| Unemployed and not currently looking for work | 6.25% |
| **Age of Children*** | |
| Under one years old | 43.75% |
| Between one and two years old | 50.00% |
| Over three years old | 56.25% |

receipt and use of the LOB. Respondents who received and accessed the LOB completed questions about the dissemination of the resource, previous usage, and possible impact on confidence and decision-making when their child was unwell. Further questions focused on the design and suggested improvements. For respondents who had received but not used the LOB, the online survey explored barriers to use, and their use of other sources of support to assist them in supporting their unwell child. All respondents noted their use of health services for their child in the previous year, as well as during the Covid-19 pandemic.

## Phase two: Individual and group interviews

At the end of the survey, respondents were asked if they would be willing to take part in either an individual or group interview and, if so, to share their contact details to arrange the interview. Individual and group interviews were offered to accommodate participant's personal preferences and to encourage a representative sample.

Those who wanted to participate in interviews were subsequently contacted by a member of the research team to provide details about this phase of the study, to provide informed consent, and to determine their preference for an individual or group interview. In total, 16 participants took part in this phase of the study, 13 through the online survey and 3 through word of mouth. The interviews were conducted by members of the research team (AJ, LS and MB) and they were audio-recorded. The responses provided by the interview participants in the Phase One survey were used to personalise the topic guide to gain a more in-depth understanding of their experiences. For instance, if the participant had indicated using the LOB for a previous

**Table 3. Individual/group interview topic guide.**

| Question | Prompts |
|---|---|
| 1.Please tell me about your experience of using the Little Orange Book | Digital / online version<br>How helpful did you find it?<br>Can you give any specific examples? |
| 2. How does / has the LOB influence(d) your decision making regarding your child's symptoms? | How did it impact on your confidence in managing symptoms?<br>How did it influence your use of health care services? |
| 3. Did the LOB play a role in your decision making during the Covid-19 pandemic? | Did your use of the LOB change during the pandemic? |
| 4. What do you think of the design of the LOB? | Can you give some examples of what you like / feel could be improved?<br>Do you have a preference for a particular format (hard copy, digital, mobile app? |
| 5. Is there anything you would like to add that we haven't discussed in relation to the LOB? | |

illness, this was directly referenced and asked about in the interview guide. The topic guide is shown below in Table 3:

The topic guide was used to focus the interviews and ensure all participants were asked the same questions to reduce interviewer bias. The aim of the interviews was to obtain a more detailed insight into the views and experiences of parents/carers regarding the role of the LOB in guiding decisions about managing childhood symptoms as well as to understand their views about the design of the resource.

## Data analysis

Descriptive statistics were calculated from the responses from the phase one online survey and are presented alongside the qualitative analysis. Open-response questions from the survey and the phase two qualitative interviews were analysed using deductive thematic analysis [13]. This involved member of the research team independently reading and re-reading the qualitative data in order to generate initial codes which were combined into broader themes. At this point, the research team met to collectively review the initial codes and themes before refining them further and agreeing on the final themes presented in the results section.

## Results

Three themes were generated from the individual interviews and the online survey open-question responses and are outlined below.

### Increasing parental empowerment in managing their child's health

This theme describes the impact of the LOB in terms of parent's empowerment in managing symptoms of childhood illness. In doing so, this provides examples of when the LOB was used, the impact on parent's confidence and the use of health services.

76.42% of survey respondents who used the LOB stated that this guided healthcare decisions when their child was unwell. To many parents/carers, the LOB was seen as a first step in identifying possible illnesses and what actions could be taken to support their child. In this sense, it was seen as valuable, quick, always available, and prevented unnecessary use of services:

*". . . they're a bit off, and I'm a bit worried about them, but I don't want to just start panicking or ringing 111 or whatever. And that's usually when we go to the orange book. . . it provides me with what to do next. . . It's quicker than ringing up 111. It's quicker than ringing up your GP. I think it's just, like, almost like a little, like, flow or how-to guide of what to do next."* (P006, Interview Participant)

First-time parents appeared to value having the LOB as a health resource, particularly those without access to support networks or concerns surrounding receiving outdated advice. Nearly half (41.78%) of survey participants who had used the LOB were first-time parents.

*". . .I thought was brilliant idea and especially as a first-time mom. . .I don't have my parents, so I can't ask them about things. I can ask my grandma, but obviously she's older. . . it's more of like old fashioned values and it might not be up to date with what's recommended to do now. . . even just to learn about them without scaring yourself"* (P015, Interview Participant)

For parents with multiple children, the LOB could be seen as less useful due to their experience and knowledge already developed with their older children, however this was unclear in both the survey and the interviews. One exception to this could be parents who had children with a large age gap.

*"It doesn't contain anything of use to me, I got it when I had my third child."* (P005, Survey Respondent)

Most (85.37%) survey respondents who had used the LOB reported increased confidence in supporting their child who was experiencing symptoms of illness. Similarly, 89.03% of survey respondents reported that the LOB had facilitated identifying the most appropriate service to assist their child. This was further reflected in the qualitative analysis, with some parents reporting using the LOB guided their use of health care services, facilitated in judging the severity of symptoms, and in some cases, directed them to the appropriate administration of medication.

*"When my little girl had chicken pox, I felt really worried, the book put my mind at rest, and I got help from the pharmacy. I bought calamine lotion, I might have given Ibuprofen, but the book advised paracetamol, I shared this advice with friends as it's not something I knew about."* (P027, Survey Respondent)

Some parents described using the information within the LOB to guide decisions surrounding schools and nursery settings. However, some parents reported the advice may not align with the procedures at nurseries.

*"I suspect different nurseries will have different policies. . . I think I remember thinking that what was considered green, amber and red wasn't necessarily what our nursery was. . . Because, for some of them you can say can you be off nursery or not. . . Yes/No. And I think our nursery. . . I felt like was taking more of a blanket kind of. . . You can come in if you've got a cold, but that was kind of about it."* (P011, Interview Participant)

Other participants described negative experiences or views which could impede usage and reduce their perception of the LOB.

*"I felt that is strongly discouraged seeking NHS assistance and that alone caused concern. I'll make that judgement myself as a parent. . . . I felt it discouraged accessing services at all"* *(P032, Survey Participant)*

*"My child burnt themselves and as per the little orange book, I asked my local pharmacist for advice. Their reaction appeared to be of surprise and why was I asking them. I felt that it was pointless and made me less confident in the advice provided in the book"* *(P026, Survey Respondent)*

Whilst the content of the LOB was seen as valuable, some participants argued the importance of also using parental instincts regarding their child's health.

*"I wouldn't say, if something in the book said, 'you don't need to seek medical advice' but something in me was telling me I needed to, I still would."* *(P014, Interview Participant)*

## Equality of access to health-based literature

The Equality of Access to Health-based Literature explores the dissemination and visibility of the LOB and the value of explanations when receiving this resource. The accessibility of health literature is directly associated with awareness of the resource including how the information is disseminated. As can be seen from Table 4, most survey respondents received the LOB at a health appointment in the community.

Some participants felt that the LOB was not easily accessible, lacked visibility, or its importance was not made clear to them on receipt.

*"More focus on it with [health visitor]. I didn't realise I had it for ages. It wasn't explained to me at all. I found it with some handouts."* *(P096, Survey Respondent)*

Conversely, one participant reported that the LOB was highly visible which could indicate variation between locations.

*". . . I'd said, oh, I felt like I'd been offered this book quite a few times–I don't know how you missed it"* *(P004, Interview Participant)*

Although an online version of the LOB is available, several participants were unaware of its online availability. Some described how this knowledge would have impacted their use of the LOB and how they would value the resource.

*"I would have downloaded it to my phone and just used it all the time. . . I think it's good to have the. . . The paper version of it, but I probably would have used the digital one. . . it's just*

**Table 4. Reports of where survey respondents received the LOB (n = 106).**

| | |
|---|---|
| **Health appointment in the community (e.g. GP surgery, midwife or health visitor)** | 64.15% |
| **Educational setting (e.g. school, nursery or childcare setting)** | 8.49% |
| **Health appointment in secondary care (e.g. Outpatients or Accident and Emergency)** | 7.55% |
| **Family member or friend** | 7.55% |
| **Accessed online** | 4.72% |
| **Community centre** | 1.89% |
| **Can't remember** | 3.77% |
| **Other** | 1.89% |

*one less thing to have to remember to pack, because you've always got on you. . . But now I know that this exists, I'll probably get my Mum to put it on her phone as well" (P009, Interview Participant)*

Some participants provided suggestions to increase the awareness and accessibility of the LOB which included providing the resource with pre-existing resources, such as "The Little Red Book" (personal child health record). Providing the LOB antenatally was seen as beneficial by some participants to allow familiarisation with the resource in advance of their child's birth.

*"Making aware of it to the mum before the baby was born, no time to consult it during the first month" (P101, Survey Respondent).*

Nearly half (47.56%) of survey respondents reported that they did not receive an explanation of the resource when it was provided to them. Crucially, interview participants highlighted that they would benefit from an explanation of the resource by health and social care staff upon receipt to support awareness and understanding.

*"I didn't know that we'd received it. . . I was sorting out some paperwork in the bookcase and the pocket that the health visitor had given us–I'd looked through some of it, but not all of it– she didn't really say what was in there. And I found it in there and realised what it was and how important it was. . . It was just 'Here's a pocket full of some information, if you need it. Have a look through and you can get in touch with us if you've got any questions.' But if I'd known what the content of it was and how important it was, I wouldn't have just. . ." (P009, Interview Participant)*

However, those who had been provided with an explanation found it beneficial.

*"My regular health visitor. . . She provided it along with a few, like, other things. . . and she talked me through it. . . You know, it's quite clear. . . This is you can manage at home. This is you might want to consult someone. And this is an emergency, you know. As well as the helpful tips it gives as well, from time to time. . . (P002, Interview Participant)*

Furthermore, 45% of respondents who had not received the LOB felt distribution via health professionals (such as midwives, general practitioners, and health visitors) would be preferable. This can also provide the ability to be reviewed during key points of contact.

*"It's provided to the parents without much explanation because they're not gonna take them when they've got a newborn, and they're gonna think that actually, that doesn't affect my child. But then to review the information of the Little Orange Book at the development checks, so such as some people have them at like 3 months, six months or a year." (P015, Interview Participant)*

## Barriers and facilitators to using the Little Orange Book

The final theme identifies the value of peer-to-peer dissemination, credibility and trustworthiness, and the clear format of information provided. Key barriers included a lack of diversity and the perception that the sole purpose of the LOB was to prevent access to health services. Recommendation of the book from peers was both a facilitator to use and dissemination. The majority of survey respondents (92.68%) who had received and used the LOB reported that

they would recommend it to others. Several participants described sharing the LOB with family members caring for their child or with friends. As a result of peer-to-peer dissemination, the resource was accessed and used more widely beyond the geographically intended audience.

> *"I have downloaded the digital format and passed it on to numerous friends who have newborns. It's very reassuring to have this to hand rather than having to rely on internet search which always seemed to provide a worst-case scenario and panic" (P014, Survey Participant)*

In contrast, only 58.33% of the 24 survey respondents who had received but not used the resource would recommend it to others.

Of the 106 survey respondents who received and accessed the LOB, 76.42% used it when their child/ren were unwell. The affiliation of the resource with the NHS was viewed positively and felt *"more reliable than searching online for answers"* (P053, Survey Respondent). However, some participants queried whether the content was maintained to ensure it was current and remained accurate while some were unaware of multiple and/or newer versions of the resource.

> *"Because it is. . . You know, medically, it's come from the NHS. . . It's come from a reliable source. And that's the information that you would get, probably, first if you rang 111. So, yeah. . . I've got masses of confidence in the orange book." (P001, Interview Participant)*

Conversely, despite acknowledging the value of the LOB, one participant queried whether the "tone" of the resource was intended to act as a deterrent from access to emergency services.

> *". . .I felt like the whole concept of the book was potentially to stop people going to A&E. And I kind of found it useful, otherwise. . . Because that's the kind of tone of the NHS, kind of, PR and stuff at the minute. The don't go to A&E with this. And don't visit your GP with that. And they're kind of pushing further on to pharmacists. I thought that was probably the essence of the book in the first place." (P004, Interview participant)*

Several features of the LOB design supported the use of the resource. 87% of participants reported that they were guided by the traffic light system to guide decision-making, including for acute symptom management in the presence of long-term conditions. 92.69% of survey respondents who had used the LOB reported the design to be useful including having tabs to differentiate the sections, although some participants suggested progression of the design to include physical tabs to support content navigation.

> *". . .the colours are really helpful and that kind of traffic light. . . system is really helpful. It's consistent throughout and it helps with that kind of accessibility that I mentioned and that you know, I guess even if the text is a challenge to you, you can see by the colour system like how alarming something is or is not, which is really positive" (P016, Interview Participant)*

Furthermore, 82.93% of respondents felt guided by the imagery within the book, finding the visual content beneficial for symptom identification and management, which was reinforced by qualitative responses and interviews. However, 47.56% would have preferred additional photographic content, which some participants acknowledged was available on the NHS website.

Several participants raised concerns about a lack of inclusivity and diversity of representation within the images, which could influence recommendations of the resource. A key

example was the meningitis rash looking visually different on different skin tones and as a result, could be "*life-threatening*" (P132, Survey respondent).

> "*The pictures are alright. . . but I think they should use babies of different races in here. Have a picture of an Asian child, a French. . .. if you can include fair and brown kids with rashes, to indicate also how does it look like. . . Instead of fair skin and then the redness of it. . .*" (P008, Interview Participant)

Of the survey respondents who had received and used the LOB, 92.60% felt that the information was easy to understand. Participants reported that the information was "*simply written*" (P006 Survey respondent), not too "*medically complex*" (P036, Survey Respondent) and "*concise and relevant*" (P007, Interview Participant). Conversely, some participants critiqued the amount of information provided but some appreciated the delicate balance with providing detailed information and the size and usability of the resource.

> "*. . .I wouldn't like to see the book being too bulky, if I'm honest, because it is a quick reference guide. And I think that's what it should be used for. But I do think there could be a little bit more context to it.*" (P003, Interview Participant)

Participants had varied preferences regarding whether the book should be available as a hard copy, digital version (online) or as a digital application (app). For some participants, a mobile app would be more accessible and useful however others valued a paper-based resource.

> "*There's something about having a hard copy in your hand there, particularly if you've got a little one there you know, and you're dealing with a screaming child and you're kind of stressed. . .* (P010, Interview Participant)

Additional comments related to the design, format and content included that the resource should signpost to other resources, such as mental health support services and paediatric first-aid training.

## Discussion and recommendations for practice

The findings of this study support assertions that parents have a range of information needs with regarding their child's health and healthcare. Parents described a sense-making process in which feelings of stress and anxiety were present but highlighted the potential of health information to reassure and signpost to available services. In doing so, the LOB was seen as valuable in supporting parental and carer decision-making about managing common childhood symptoms of illness. Many participants reported the LOB had increased their knowledge of common complaints but also helped them to decide whether there was a need to contact healthcare services; previous research has suggested that these are two key factors in increasing the appropriate utilisation of health services [1, 7, 8]. This aligns with the purpose of the LOB which was intended to help parents and carers use services appropriately. The LOB was found to particularly benefit new parents and those with an age gap between their children; expectant and new parents have been identified as having specific educational needs which include a need to understand 'what is normal' and reduce levels of anxiety [13]. Here, this suggests there is a real need for information and support during these key stages.

Some participants reported trusting the LOB due to this being an NHS resource. However, there was also a need to enhance the resource, particularly in relation to improving inclusivity

but also consistency. There were examples where other services, such as pharmacies and nurseries, had different practices from those outlined in the LOB. This is a key challenge for both healthcare professionals and the recipients of advice and is known to be increasingly important as the sources of information about health increase [14]. Whilst it is difficult to understand the full impact of receiving conflicting information, it is known that this can lead to confusion as well as criticism and mistrust of services [14] and highlights the need for joined-up policy and information practices. The consequences may be linked to levels of health literacy and empowerment and so may affect some service users more than others. What must be noted is that the sample included in this evaluation may not be reflective of the wider population as 80% of participants were in employment and 65.86% were educated to degree level or above. This could suggest a link between educational attainment and increased likelihood of using the LOB, as well as levels of confidence and ability to use a range of sources of information to make appropriate decisions. Wider literature notes the well-established link between education and health and the widening health gap between those who are less educated than their peers [15]. Although educational attainment does not necessarily mean in-depth knowledge of child health, it does increase the ability to access, understand and use healthcare information [16]. Therefore, it is important that work is done to ensure that all families have access to relevant information to care for their children. This should incorporate a range of different approaches including resources such as the LOB and materials in both hard copy and digital formats, which was noted as beneficial in this evaluation, as well as the support of key services.

Consideration should be given to how resources such as the LOB are distributed but also how parents are guided in their use. This evaluation suggested that universal services, such as midwives and health visitors, are well placed to distribute hard copies of the LOB and raise awareness of digital access, and that these interactions may be valuable opportunities for healthcare professionals to signpost parents. This connects well with their role in supporting children and families. Indeed, in recent years there has been renewed political emphasis on the importance of supporting parents/carers during the first 1001 days of life and governmental ambitions to provide joined-up support, as well as information for families when needed [17]. It must be noted that these ambitions are set against a backdrop of challenges for the health and social care sector in relation to long-term funding and staff shortages. Of particular note is the impact on key services, such as health visiting, which are tasked with delivering this agenda in the presence of immense challenges for parents and families, including the cost-of-living crisis, increased numbers of parents living with mental health problems and increased prevalence of domestic abuse and adversity, all posing real risks to the health and wellbeing of babies and young children [18]. Alongside this, there has been increasing recognition that there are insufficient health visitors to meet the needs of families with a loss of almost 40% of the health visiting workforce since 2015 [18]. This is perhaps where resources such as the LOB play an important role. Indeed, this evaluation found that during the pandemic, the LOB was valuable in the context of limited access to services and the message to stay home, however some parents felt strongly that the LOB was a mechanism to deter the use of services. This underlines the need for the LOB to be explained and used to compliment, rather than replace other forms of support.

A key strength of this evaluation is that the overwhelming support for the LOB as an intervention was consistent with findings of a previous evaluation, which included professionals and service users [9]. This suggests that there is a real need for educational and decision-making resources, although as mentioned previously this is perhaps reflective of the high levels of educational attainment in the sample. The mixed methods approach allowed us to capture the views of 128 participants and to ask in-depth questions about the use of the LOB. Whilst the respondents were overwhelmingly mothers, the sample did include some fathers, foster carers,

adoptive parents and parents of minority ethnic origin. A larger sample still might increase diversity and provide more insight into how the LOB could be developed and meet the needs of more parents/carers.

Resources such as the LOB can provide important information to support parents and carers in decision-making. However, this evaluation shows that careful thought should be given to how they are developed, delivered and integrated with other local services to have wider benefit to all parents and carers. This includes reviewing how inclusive the content is but also conducting evaluations such as this to determine who is using the resource and how, and identifying groups who may not be accessing appropriate information resources but have support needs with regards to making decisions about children's healthcare. The overwhelming support for the LOB suggests that it needs to be more widely embedded across Newcastle and Gateshead but equally that the resource is likely to be of benefit to parents and carers beyond these locations.

## Supporting information

**S1 File. Survey.**
(DOCX)

## Acknowledgments

With thanks to the parents and carers who took part in this evaluation.

## Author Contributions

**Conceptualization:** Amy Johnson, Kathryn Carruthers, Matthew Breckons, Lynette Harland Shotton.

**Data curation:** Matthew Breckons.

**Formal analysis:** Amy Johnson, Matthew Breckons, Lynette Harland Shotton.

**Funding acquisition:** Kathryn Carruthers, Matthew Breckons, Lynette Harland Shotton.

**Methodology:** Amy Johnson, Kathryn Carruthers, Matthew Breckons.

**Project administration:** Lynette Harland Shotton.

**Resources:** Lynette Harland Shotton.

**Supervision:** Lynette Harland Shotton.

**Writing – original draft:** Amy Johnson, Kathryn Carruthers.

**Writing – review & editing:** Kathryn Carruthers, Matthew Breckons, Lynette Harland Shotton.

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
