## [Decision Letter · Decision Letter 0]

11 Mar 2024

PONE-D-23-37634“The impact of the Little Orange Book on how parents/carers manage symptoms of illness in children: A mixed methods study”.PLOS ONE

Dear Dr. Shotton,

Thank you for submitting your manuscript to PLOS ONE. After careful consideration, we feel that it has merit but does not fully meet PLOS ONE’s publication criteria as it currently stands. Therefore, we invite you to submit a revised version of the manuscript that addresses the points raised during the review process.

We look forward to receiving your revised manuscript.

Kind regards,

Cari Malcolm

Academic Editor

PLOS ONE

 [Grant Number Agresso Reference CRP01901

L H Shotton – grant holder

Funder - Newcastle Gateshead CCG, 

Riverside House, 

Goldcrest Way, 

Newburn Riverside (Business Park), 

Newcastle upon Tyne 

NE15 8NY

https://nenc-newcastlegateshead.icb.nhs.uk

A J received salary 

There are no competing interests.].  

Reviewers' comments:

Reviewer's Responses to Questions

**Comments to the Author**

1. Is the manuscript technically sound, and do the data support the conclusions?

Reviewer #1: Partly

Reviewer #2: Partly

2. Has the statistical analysis been performed appropriately and rigorously? 

Reviewer #1: I Don't Know

Reviewer #2: No

3. Have the authors made all data underlying the findings in their manuscript fully available?

Reviewer #1: No

Reviewer #2: No

4. Is the manuscript presented in an intelligible fashion and written in standard English?

Reviewer #1: Yes

Reviewer #2: Yes

5. Review Comments to the Author

Reviewer #1: Thank you for allowing me to read your article, this is an important and current topic and your research is needed. I have also downloaded and looked at the Little Orange Book and really enjoyed it. Here are my views on your article.

Generally:

- Small typos and missed letters, please proofread your text.

- Doesn’t appear like you used reporting guidelines. I don't know if there are any specific guidelines for mixed methods, but for example SRQR or COREQ are suitable to help improve the transparency of all aspects of the qualitative part of your research.

- In several places throughout the article Covid 19 is mentioned, but it is sparingly described and feels a little taken out of context. As a reader, I struggle a bit with

Title

- Good title that reflects what your article is about

Abstract

Objective: is not the same as what is written in the introduction, nor is it the aim of your study

Method: Thinking about whether it really is perceptions you’re seeking, perhaps rather experiences?

Results: There are not 128 surveys included in your results, nor 16 interviews, I would suggest writing the actual numbers here and maybe include it under method and not under results.

Introduction

Lines 41-47: Good entry into the introduction

You highlight relevant topics in the introduction, however, as a reader I am having some problems following along in the text. For example; the first sentence on line 50 feels cut off and taken out of context. Overall, the common thread is missing and it becomes difficult to see how everything is connected. I would suggest that you work on this text more to find a flow and funnel down towards your problem formulation. Right now, I am having difficulties understanding what the problem is and where your research fits.

Objective/ aim

You describe a primary aim, is there a secondary aim?

Your aim is long and detailed, I would suggest a more comprehensive aim, for example: The aim of this study was to understand parent/carer views (experiences?) of using the LOB.

The rest of what is described as aim could belong to the overall description of the problem.

Design

Line 74: Perhaps you could clarify that these two phases consist of a quantitative part and a qualitative part.

Lines 78-82: Starting with the sentence beginning with "While the survey...". This information should preferably be placed design in the introduction, or think about how this information is relevant in relation to this study.

Lines 82-83. I would prefer that ethics have its own heading. I would also like some more information regarding ethics such as whether you obtained consent and how you established confidentiality.

Participants

Criterias for inclusion/exclusion are lacking. You excluded everyone who did not receive the book, then I am wondering why they were allowed to answer the survey? Your aim is to evaluate the use of the book, it would make more sense that only those who received the book should answer the survey? That is something you could explain in the discussion.

I would like to know which sampling methods were used and how you calculated your sample size, and these are also things that you can discuss later in the paper.

Tables: Hard to read, would have liked all the text to be in one column and the numbers to stand on their own. I suggest you remove the column with those who did not receive the book because they are not included in the result.

Ex.

Recieved and used LOB (n=82) Received and did not use LOB (n= 24)

Ethnicity

White 93.90% 3.66%

Black… 91.67% 0%

Etc.

Line 103: Here you write how many were actually included in your analysis, It would be less confusing if you use the same numbers everywhere (eg in the abstract).

Online survey

Overall: I would have liked more information about your survey. For example; which one have you used, how is it created, is it tested and validated?

One suggestion is that you attach the survey or put the questions in a box. If the survey was made specifically for this study, please clarify this.

Line 115: that your participants could win money for their participation should preferably be placed in the ethics section.

Interviews

You have chosen to do both individual and group interviews, why? Here you also change the method when you include three people via word of mouth, why? In general, I am having trouble understanding how your interviews/group interviews were conducted, please attach your interview guide. This is also something you should consider including in your method-discussion and clarify how it may have affected the results.

Data analysis

This section needs to be developed, as it is now it is not possible for me as a reader to understand how and what you have done. The analysis process needs to be described. The descriptive analysus part also needs to be developed, eg what measures were used and were there any internal loss?

Results

It is difficult to assess the trustworthiness of the results because the data-analysis is not properly described. In general; the result is not perceived as fully analyzed. There are repetitions and similar results are under several headings. The quotes get more space than the result itself.

I would have liked a table to illustrate the quantitative results, especially since it’s not specified in the method section which questions were asked.

Discussion and recommendations for practice

Could be divided under two headings for easier reading.

Your discussion is interesting and well-formulated, perhaps I would have liked the discussion to be based on more references.

Missing a clear method discussion that includes concepts such as trustworthiness; dependability, confirmability, credibility, transferability (qualitative research) and validity and reliability (quantitative research).

You write a lot about the previous evaluation of LOB, perhaps you would have benefited from clarifying everything about it in the introduction.

Line 385: It says 128 participants, but that’s not the number that were included in the results.

Lines 390-398: I interpret this as your conclusion, but it becomes a bit unclear when it is included in the discussion

Reviewer #2: Thank you for the opportunity to review this manuscript entitled The impact of the Little Orange Book on how parents/carers manage symptoms of illness in children: A mixed methods study. It addresses an important and contemporary issue and makes a contribution to the existing evidence base. There are several areas requiring attention which I have outlined in detail below:

Abstract – The abstract would benefit from being re-written. It doesn’t align well with the paper, especially the objectives.

Introduction – Some of the existing literature around this topic is referred to, albeit it is brief and doesn’t provide a clear overview.

I would suggest amending the first sentence for clarity, perhaps: ‘For parents and carers of young children, making decisions about when and where to access health care or services when they are acutely unwell is complex.’

I don’t understand what you mean by ‘research highlights difficulties in making appropriate decisions’ – are you referring to parents having difficulty making these decisions? If so, say that.

‘A recent systematic review suggested that pre-disposing factors (socioeconomic status, ethnicity and race) may influence parental decisions on the use of care which are often based on perceptions of the urgency of the condition [3]’ – These ‘pre-disposing factors’ are just a small part of the picture and this systematic review made far wider and valuable contributions to the evidence base around parental decision-making in accessing unscheduled care. I suggest re-visiting this source and providing a more comprehensive and accurate overview of the main messages.

‘Prompted by dramatic reductions in the use of children’s health services during the Covid-19 pandemic….’ If you are going to make a strong statement like this, you will need to cite the quantitative data source which tells you there were ‘dramatic reductions’ in paediatric access to health services.

‘While health information has the potential to cause anxiety [5].’ – Incomplete sentence.

‘The resource aims to support parents/carers (< 5 years of age)’ – Please re-phrase as it sounds like the parents are less than five years of age.

Please be clearer about the aim of this paper and, importantly, it differs to what was published in the evaluation. This is fundamental to satisfy the key criteria of this journal i) whether the study presented the results of original research and ii) whether the results reported have not been published elsewhere.

Methods –

How did the online survey inform the qualitative interview component – this needs better explained.

Please use an appropriate reporting tool for qualitative interview studies (ie COREQ)

Where is the interview topic guide? This needs to be included as a supplementary file.

‘All respondents noted their use of health services for their child in the previous year, as well as during the Covid-19 pandemic.’ – this requires an explanation.

Much greater detail on the process of thematic analysis is required.

Findings – Please explain how you have integrated the survey and interview findings – without a clearer explanation of your methods earlier, it is difficult to follow the narrative of your findings and, more importantly, to have confidence in the trustworthiness of the data.

Further in-depth analysis is needed for all themes. The findings are very descriptive with little text and an overuse of exemplar quotations.

Discussion – The discussion is interesting and with further integration of the literature allowing comparison and contrasts with your findings and the existing evidence base, it would be much stronger. You need to further consider some of the limitation in your work.

6. PLOS authors have the option to publish the peer review history of their article (what does this mean?). If published, this will include your full peer review and any attached files.

Reviewer #1: **Yes: **Emma Westin

Reviewer #2: No

---

## [Author Response · Author response to Decision Letter 0]

16 Sep 2024

Dear Reviewers,

thank you very much for your detailed and constructive feedback. We have reviewed each point and have uploaded a table which outlines how we have responded to each comment. We hope this revised submission is now acceptable and look forward to a decision in due course.

---

## [Editor Report · Decision Letter 1]

29 Sep 2024

“The impact of the Little Orange Book on how parents/carers manage symptoms of illness in children: A mixed methods study”.

PONE-D-23-37634R1

Dear Lynette

We are pleased to inform you that your manuscript has been judged scientifically suitable for publication and will be formally accepted for publication once it meets all outstanding technical requirements.

Kind regards,

Cari Malcolm

Academic Editor

PLOS ONE
---

## [Editor Report · Acceptance letter]

8 Oct 2024

PONE-D-23-37634R1 

PLOS ONE

Dear Dr. Shotton, 

I'm pleased to inform you that your manuscript has been deemed suitable for publication in PLOS ONE. Congratulations! Your manuscript is now being handed over to our production team.

Kind regards, 

on behalf of

Dr. Cari Malcolm 

Academic Editor

PLOS ONE